# Multimodal Federated Learning via Contrastive Representation Ensemble

**Qiying Yu**[1,4], **Yang Liu**[1,4*], **Yimu Wang**[2], **Ke Xu**[3], **Jingjing Liu**[1*]
[1] Institute for AI Industry Research, Tsinghua University
[2] University of Waterloo    [3] Carnegie Mellon University
[4] Shanghai Artificial Intelligence Laboratory
`yuqy22@mails.tsinghua.edu.cn`, `{liuy03,jjliu}@air.tsinghua.edu.cn`

## Abstract

With the increasing amount of multimedia data on modern mobile systems and IoT infrastructures, harnessing these rich multimodal data without breaching user privacy becomes a critical issue. Federated learning (FL) serves as a privacy-conscious alternative to centralized machine learning. However, existing FL methods extended to multimodal data all rely on model aggregation on single modality level, which restrains the server and clients to have identical model architecture for each modality. This limits the global model in terms of both model complexity and data capacity, let alone task diversity. In this work, we propose *Contrastive Representation Ensemble and Aggregation for Multimodal FL (CreamFL)*, a multimodal federated learning framework that enables training larger server models from clients with heterogeneous model architectures and data modalities, while only communicating knowledge on public dataset. To achieve better multimodal representation fusion, we design a global-local cross-modal ensemble strategy to aggregate client representations. To mitigate local model drift caused by two unprecedented heterogeneous factors stemming from multimodal discrepancy (*modality gap* and *task gap*), we further propose inter-modal and intra-modal contrasts to regularize local training, which complements information of the absent modality for uni-modal clients and regularizes local clients to head towards global consensus. Thorough evaluations and ablation studies on image-text retrieval and VQA tasks showcase the superiority of CreamFL over state-of-the-art FL methods.

## 1 Introduction

Federated Learning (FL) (Yang et al., 2019; Li et al., 2020; Kairouz et al., 2021; Zhao et al., 2018), a decentralized training paradigm that allows multiple parties to collaboratively train models without compromising privacy, has emerged as an alternative to centralized machine learning. Most existing FL methods only consider scenarios where the private data from clients belong to the same modality (*e.g.*, image or text). However, with the fast development of mobile technology and IoT infrastructures (Brunete et al., 2021) that harness data from different modalities (*e.g.* sensory, visual, audio) with privacy constraints, there is an increasing need for advanced FL algorithms to allow the training of larger and capable model that can absorb heterogeneous private data (across modalities) at edge and simultaneously handle diverse multimodal tasks (Gan et al., 2022; Chen et al., 2020b).

In the past, there has been some early attempts at applying FL to multimodal tasks (Xiong et al., 2022; Zhao et al., 2022; Liu et al., 2020), which all adopt the FedAvg (McMahan et al., 2017) framework by using homogeneous models for each modality. In practice, however, edge devices may have limited computational and memory resources, restraining the capacity of the global model to smaller and lighter scales. Moreover, naive aggregation of modality-dependent models is inadequate in addressing the model drift (Karimireddy et al., 2020) problem between clients.

Recently, a few algorithms (Cho et al., 2022; Cheng et al., 2021) have been proposed to enable larger server model training. For example, FedET (Cho et al., 2022) proposes an ensemble Knowledge Distillation (KD) based framework to enable a large model at server and relatively small yet

---
*Corresponding Authors

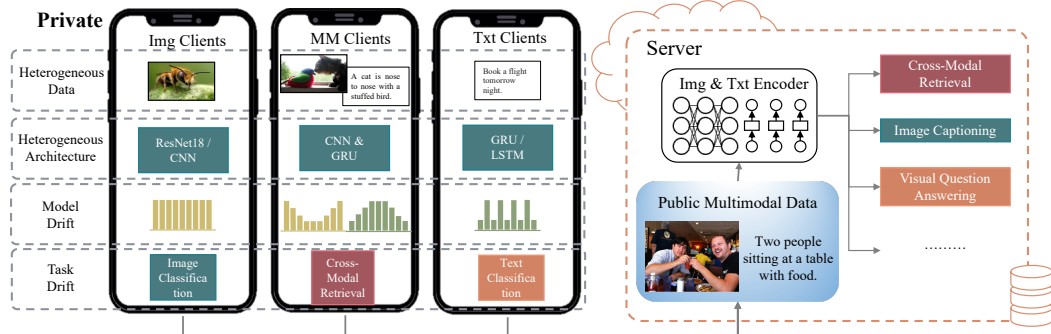

Figure 1: Illustration of multimodal FL. A large model at server supports multimodal tasks with public data, and heterogeneous clients at edge handle uni- and multi-modal tasks with private data.

deployable models on edge devices. However, they transfer and ensemble knowledge from a bag of client teachers through *logit*, which is difficult to extend to multimodal setting. Most multimodal tasks (e.g., image/video captioning (Vinyals et al., 2015)) typically operate on fused cross-modality representation level, whereas existing strategies for aggregating logits are no longer applicable.

In this paper, we design a novel KD-based multimodal federated learning framework, *CreamFL (Contrastive Representation Ensemble and Aggregation for Multimodal FL)*, which simultaneously leverages uni- and multi-modal data across heterogeneous clients to learn a larger global model, through representation-level ensemble knowledge transfer. The global model learns from clients via communicating private knowledge on public dataset from diverse client networks without revealing private models and data. CreamFL transmits *low-dimensional representations* of public data between server and clients, which are usually contextual and applicable to more complex tasks than logits. To effectively aggregate representations transmitted from heterogeneous clients, we propose a *global-local cross-modal contrastive* aggregation strategy, to 1) filter out drifting outliers by contrasting *local* representations to *global* ones; 2) pick out outstanding candidates that better match their paired partners, by contrasting to representations from another modality.

Moreover, FL with multimodal data brings about two new types of model gap: *modality gap* and *task gap*. Uni-modal clients trained under a single modality (*e.g.* image) have never seen data from other modalities (*e.g.*, text) in the training procedure, therefore lacking the capability of recognizing another modality. We call this mutual incompatibility between clients the 'modality gap'. Task gap refers to the fact that different clients may be trained for diverse tasks, *e.g.*, uni-modal clients for image classification task and multimodal clients for image-text retrieval task. Both gaps cause unprecedented model drift (Karimireddy et al., 2020) problems. To tackle this, we introduce two novel contrastive objectives to regularize local training. An *inter-modal contrastive objective* is designed to mitigate the modality gap, by performing cross-modality contrasts using public data in the local training phase, which complements for the information of the absent modality in uni-modal clients. To bridge the task gap, an *intra-modal contrastive objective* is proposed to contrast local representations to their corresponding global ones in each modality, regularizing models to head towards the global consensus (Li & Wang, 2019).

In summary, 1) CreamFL is the first KD-based multimodal FL framework to support heterogeneous modality and model architectures between server and clients, while only communicating private knowledge on public dataset without revealing private models and data. Experiments show that CreamFL outperforms other FL systems in multimodal setting in terms of both model performance and communication cost; 2) CreamFL ensembles representations instead of logits for knowledge transfer between clients and server, with a novel global-local cross-modal aggregation strategy for better representation learning and inter/intra-modal contrastive objectives to address model drift; 3) Our framework enables larger model training at server that absorbs modality-diverse knowledge from resource-constrained clients, which is required for complex cross-modal tasks.

## 2    RELATED WORK

Some works investigated applying KD to FL (Lin et al., 2020; Itahara et al., 2021; Wu et al., 2021). Li & Wang (2019) allows heterogeneous clients to distill knowledge from the aggregated consensus,

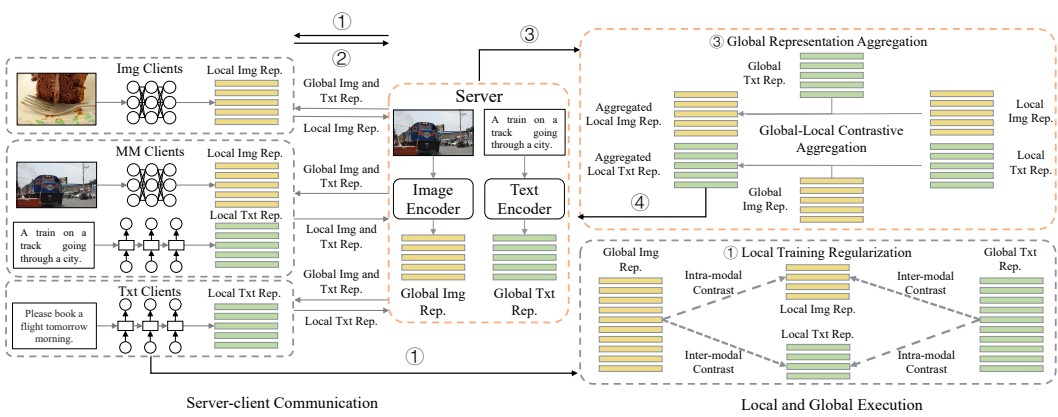

Figure 2: Illustration of CreamFL framework with a large server and heterogeneous clients. ①
Clients receive global representations (Global Img/Txt Rep.) from the server and perform regularized local training. ② Clients generate representations of public data (Local Img/Txt Rep.) and transmit them to the server. ③ The server aggregates received local representations with global representations. ④ The server distills knowledge from the aggregated representations.

but it does not train a server model. FedGKT (He et al., 2020a) allows larger server models, while the server makes no selection among clients, leading to a bad consensus with impeded performance. Cho et al. (2022) and Cheng et al. (2021) concurrently propose to train larger server models in FL through ensemble knowledge transfer. However, their selective aggregation strategies are specifically designed for logit (using variance or entropy of logit for weighting) and the learned large server model is limited to classification task. We propose to transmit *representations* and design a novel ensemble strategy to make the model applicable to complex multimodal tasks.

Several attempts have been made to apply FL to multimodal tasks. Xiong et al. (2022) extended FedAvg to multimodal scenario. Liu et al. (2020) applied FL to leverage multiple datasets from different distributions to boost performance, but lacks the communication phase of FL. Zhao et al. (2022) proposed to assign higher weights to multimodal clients in aggregation, but this strategy needs manual tuning for the weights and only works for FL systems with both uni- and multimodal clients. We design a novel tuning-free global-local cross-modal contrastive strategy for aggregation, which has broader applicability to scenarios with only multimodal clients.

Inter-modal contrastive learning is a prevailing technique for multimodal self-supervised learning (Jia et al., 2021; Singh et al., 2022; Yuan et al., 2021). The contrastive part of most pretraining methods resembles that of CLIP (Radford et al., 2021), where aligned and unaligned image-text pairs are treated as positive and negative pairs, respectively. Our method differs from them in two aspects: 1) we creatively apply the concept of contrasting to design a "metric" for evaluating the quality of representations, to address the challenge that the representation quality varies significantly between clients due to their inevitable heterogeneity; 2) we propose to use inter-modal contrast as a regularization technique to bridge the modality gap between clients, and restrict global-local discrepancy.

## 3 FEDERATED MULTIMODAL LEARNING

A key goal of multimodal learning is to unify signals from different modalities into the same vector space, where semantically correlated data across modalities share similar representations. This goal serves as a north star across our design of the multimodal federated learning framework, to guide the global model to better capture intrinsic correlation between multimodal data.

### 3.1 PROBLEM DEFINITION

We consider a heterogeneous FL setting where $M$ multimodal clients, $I$ uni-modal image clients and $T$ uni-modal text clients collaboratively train a global model $f_s(\cdot; \mathbf{w}) : \mathbb{R}^n \to \mathbb{R}^d$ through representation-level ensemble knowledge distillation, where $\mathbf{w}$ is the model parameter, $n$ and $d$ are

the dimensions of input data and extracted features of the input data, respectively. Each image client $p \in [I]$ has its own private dataset $\mathcal{I}_p = \left\{ \left( x_p^k, y_p^k \right) \right\}_{k=1}^{|\mathcal{I}_p|}$, where $x_p^k$ is the $k$-th training sample of the $p$-th image client, $y_p^k$ is its corresponding label. Similarly, each text client $q \in [T]$ has its private dataset $\mathcal{T}_q = \left\{ \left( x_q^k, y_q^k \right) \right\}_{k=1}^{|\mathcal{T}_q|}$. The multimodal client $m \in [M]$ has its local dataset $\mathcal{M}_m = \left\{ \left( i_m^k, t_m^k \right) \right\}_{k=1}^{|\mathcal{M}_m|}$, where $(i_m^k, t_m^k)$ is $k$-th image-text pair with correlated semantic information. Both server and clients can access a public dataset $\mathcal{P} = \left\{ \left( i^{(k)}, t^{(k)} \right) \right\}_{k=1}^{|\mathcal{P}|}$, where $\left( i^{(k)}, t^{(k)} \right)$ is an image-text pair. We further assume that each client adopts a small model $f_c(\cdot; \mathbf{w}_c) : \mathbb{R}^n \to \mathbb{R}^d$ of its own choice based on its local data, task and resources, which outputs representations of the same dimension $d$ as the global model.

As shown in Figure 2, during training, clients first receive global representations of public data and perform multiple local steps of representation learning with inter- and intra-modal regularization (section 3.2). Then, clients generate representations of public data according to their own modality and transmit them to the server. To eliminate bias between different modalities and clients, the server selectively aggregates representations (section 3.3). Finally, the server performs knowledge distillation from the aggregated representations and transmits its trained representations back to clients. The complete algorithm is provided in Algorithm 1.

## 3.2 LOCAL TRAINING VIA CONTRASTIVE REGULARIZATION (LCR)

We first focus on mitigating model drift through regularizing local training, by designing contrastive objectives from two perspectives: 1) *inter-modal contrast* to complement the absent modality information for uni-modal clients and enhance cross-modal interaction; and 2) *intra-modal contrast* to mitigate non-i.i.d. problem in each modality separately. At the beginning of each communication round, the server sends the representations of public dataset generated by the global model to clients to help them regularize local training ('Global Img Rep.' and 'Global Text Rep.' in the '① Local Training Regularization' path of Figure 2).

### 3.2.1 INTER-MODAL CONTRAST

Modality gap exists between uni-modal and multimodal clients. For example, image clients have never seen any text information during their training process, while multimodal clients are trained under the presence of both modalities. This gap leads to the discrepancy between image representations generated by image- and multimodal-clients. To address this issue, inspired by CLIP (Radford et al., 2021) that enhances visual representation learning by contrasting to language, we perform an *inter-modal* contrast in clients' local training to complement information from the absent modality.

We take the regularization of image clients as example (similar procedure for text clients). Considering the local training of image client $c$, at the beginning of each communication round, $c$ receives the global text representations of public data $t_{\text{global}}^{(j)} \in \mathbb{R}^d, j = 1, \ldots, |\mathcal{P}|$. For the $k$-th image $i^{(k)}$, client $c$ first generates its representation $i_{\text{local}}^{(k)} \in \mathbb{R}^d$. However, current $i_{\text{local}}^{(k)}$ may be heavily biased due to: 1) local training data is non-i.i.d. sampled, similar to traditional FL; 2) local training is performed under single image modality, lacking not only the language information, but also the cross-modal alignment interaction, a unique challenge in multimodal FL.

To address this problem, we resort to global text representations to guide uni-modal image clients to head towards the unified multimodal representation space. We assume the global multimodal server has partially learned a shared representation space between image and text representations. By encouraging the coupling of $i_{\text{local}}^{(k)}$ and its paired global data $t_{\text{global}}^{(k)}$, we can enforce the uni-modal image clients to head towards the shared multimodal representation space, while keeping away from other unrelated text data points $t_{\text{global}}^{(j)}, j = 1, \ldots, |\mathcal{P}|, \neq k$. This is achieved by an inter-modal contrastive loss:

$$\ell_{\text{inter}}^{(k)} = -\log \frac{\exp\left( {i_{\text{local}}^{(k)}}^{\top} \cdot t_{\text{global}}^{(k)} \right)}{\sum_{j=1}^{|\mathcal{P}|} \exp\left( {i_{\text{local}}^{(k)}}^{\top} \cdot t_{\text{global}}^{(j)} \right)} \tag{1}$$

---

**Algorithm 1:** CreamFL algorithm.

---

**Input:** Public dataset $\mathcal{P}$, number of communication rounds $T$, number of clients $C$, number of local epochs $E$, server model $f_s$, model $f_c$, dataset $\mathcal{D}_c$ of the $c$-th client and fraction of clients $p$ that perform computation in each round.

**Output:** The final server model $\mathbf{w}^T$

---

1 **ServerExecutes**:

2     **for** $t = 0, 1, \ldots, T - 1$ **do**

3         $\boldsymbol{I}_{\text{glob}}, \boldsymbol{T}_{\text{glob}} \leftarrow f_s(\mathcal{P}; \mathbf{w}^t);$       ▷ generate global representations

4         $\boldsymbol{I}_{\text{local}}, \boldsymbol{T}_{\text{local}} \leftarrow [], [] \ ;$       ▷ sets of local representations

5         $S_t \leftarrow$ random set of $\max(p \cdot C, 1)$ clients;

6         **for** each client $c$ in $S_t$ **do**              ▷ in parallel

7             send public representations $\boldsymbol{I}_{\text{glob}}, \boldsymbol{T}_{\text{glob}}$ to client $c$ ;

8             $\boldsymbol{I}_c, \boldsymbol{T}_c \leftarrow$ **ClientLocalTraining**$(c, t, \boldsymbol{I}_{\text{glob}}, \boldsymbol{T}_{\text{glob}});$

9             $\boldsymbol{I}_{\text{local}}, \boldsymbol{T}_{\text{local}} \leftarrow \boldsymbol{I}_{\text{local}} + \boldsymbol{I}_c, \boldsymbol{T}_{\text{local}} + \boldsymbol{T}_c$ ;

10         **end**

11         Train server model $\mathbf{w}^t$ using the public multimodal dataset $\mathcal{P}$ ;

12         $\boldsymbol{I}, \boldsymbol{T} \leftarrow$ Aggregate $\boldsymbol{I}_{\text{local}}, \boldsymbol{T}_{\text{local}}$ with $\boldsymbol{I}_{\text{glob}}, \boldsymbol{T}_{\text{glob}}$ according to Equation 4-6;

13         $\mathbf{w}^{t+1} \leftarrow \mathbf{w}^t$ distills knowledge from aggregated $\boldsymbol{I}, \boldsymbol{T}$ accroding to Equation 7;

14     **end**

15 **ClientLocalTraing**$(c, t, \boldsymbol{I}_{\text{glob}}, \boldsymbol{T}_{\text{glob}})$:

16     **for** epoch $i = 0, 1, \ldots, E - 1$ **do**       ▷ regularized local training

17         $\mathbf{w}_c^{(t,i+1)} \leftarrow \mathbf{w}_c^{(t,i)} - \eta_c \nabla \mathcal{L}_{\text{local}}^{(c)}(\mathcal{D}_c, \boldsymbol{I}_{\text{glob}}, \boldsymbol{T}_{\text{glob}}; \mathbf{w}_c^{(t,i)})$ ;    ▷ Equation 1-3

18     **end**

19     $\boldsymbol{I}_c, \boldsymbol{T}_c \leftarrow f_c(\mathcal{P}; \mathbf{w}_c^{(t,E)});$       ▷ generate local representations

20     return $\boldsymbol{I}_c, \boldsymbol{T}_c$ ;

---

Through regularization by this contrastive objective, we not only inform the local image client the existence of another modality, but also regularize it towards the shared multimodal representation space to learn better cross-modal interaction.

### 3.2.2 INTRA-MODAL CONTRAST

The task gap between uni- and multimodal clients arises naturally in multimodal FL. For example, an image client is trained by image classification task while a multimodal client is trained under cross-modal retrieval, which can induce severe model drift because these clients are trained towards different targets. To alleviate such drift, we introduce an intra-modal contrast to regularize local representations towards their global consensus, in each separate modality.

Concretely, image client $c$ receives global image representations $\boldsymbol{i}_{\text{global}}^{(j)}, j = 1, \ldots, |\mathcal{P}|$ from the server at the beginning of each communication round. We contrast the representation of the $k$-th image $\boldsymbol{i}_{\text{local}}^{(k)}$ to its corresponding global version $\boldsymbol{i}_{\text{global}}^{(k)}$ to guide local model towards the global consensus. Following MOON (Li et al., 2021), we add a negative contrast between $\boldsymbol{i}_{\text{local}}^{(k)}$ and the representation generated by the local model of the last round $\boldsymbol{i}_{\text{prev}}^{(k)}$ to increase the distance between current local model and the previous one. Different from MOON, our intra-modal contrast uses public data as a bridge without operating on private data, thus can be seen as an extension of MOON to a more scalable KD-based FL framework. The intra-modal contrastive loss is defined as:

$$\ell_{\text{intra}}^{(k)} = -\log \frac{\exp\left(\boldsymbol{i}_{\text{local}}^{(k)}{}^{\top} \cdot \boldsymbol{i}_{\text{global}}^{(k)}\right)}{\exp\left(\boldsymbol{i}_{\text{local}}^{(k)}{}^{\top} \cdot \boldsymbol{i}_{\text{global}}^{(k)}\right) + \exp\left(\boldsymbol{i}_{\text{local}}^{(k)}{}^{\top} \cdot \boldsymbol{i}_{\text{prev}}^{(k)}\right)} \tag{2}$$

The final objective of client $c$ regularized by inter- and intra-modal contrasts can be written as:

$$\mathcal{L}_{\text{local}}^{(c)} = \frac{1}{|\mathcal{I}_c|} \sum_k \ell(x_c^k, y_c^k; \mathbf{w}_c) + \gamma \cdot \frac{1}{|\mathcal{P}|} \left( \sum_k \ell_{\text{inter}}^{(k)} + \sum_k \ell_{\text{intra}}^{(k)} \right) \tag{3}$$

With a slight abuse of symbols for simplicity, $\ell(x_c^k, y_c^k; \mathbf{w}_c)$ is the objective of the original local training task, $\gamma$ is the coefficient of our proposed regularization terms for mitigating local drift. After local training, clients generate representations of public data according to their own modality (*e.g.*, image client only generates image representations while multimodal clients generate both image and text representations) and transmit them to the server. The server aggregates received representations and then distills knowledge from the aggregated version.

### 3.3 GLOBAL-LOCAL CONTRASTIVE AGGREGATION (GCA)

Representation aggregation has been underexplored before, either in the area of FL or in ensemble knowledge distillation. As many representations are highly biased or even maliciously attacked, the server has to selectively aggregate them, and the key challenge is to precisely evaluate the quality of representations to decide which one should contribute more during ensemble.

In the context of multimodal learning, we answer this question by designing a global-local cross-modal contrastive score for weighting purposes. Generally, *local* representation that is "close" to its paired partner' *global* representation generated by server model and resides far away from other unpaired samples, always better captures the semantic information and cross-modal interaction of data. In the meantime, it often causes less local drift because it is benchmarked to the *global* representations that contain information from all clients.

#### 3.3.1 GLOBAL-LOCAL CROSS-MODAL CONTRAST

For the $k$-th image-text pair $\left(i^{(k)}, t^{(k)}\right)$ of public dataset $\mathcal{P}$, before aggregating the received local representations, the server first generates the global representations of public data $\boldsymbol{i}_{\text{global}}^{(j)}, \boldsymbol{t}_{\text{global}}^{(j)} \in \mathbb{R}^d, j = 1, \ldots, |\mathcal{P}|$ by the current server model ('Global Img/Txt Rep.' in the '③ Global Representation Aggregation' path of Figure 2). Taking the aggregation of the $k$-th local image representations $\boldsymbol{i}_{\text{local}}^{(k,c)}, c \in [C = I + M]$ as example, the superscript $(k,c)$ denotes it is the $k$-th image representation from the $c$-th client. Complying with the golden pursuit of multimodal learning, we assign a higher weight in aggregation to the local representation $\boldsymbol{i}_{\text{local}}^{(k,c)}$ that better matches its counterpart's global representation $\boldsymbol{t}_{\text{global}}^{(k)}$ and less approximates other texts $\boldsymbol{t}_{\text{global}}^{(j)}, j \neq k$. We achieve this by the spirit of contrastive learning (He et al., 2020b; Chen et al., 2020a), and compute the score of each local image representation in the form of contrastive loss, as follows:

$$s^{(k,c)} = \log \frac{\exp\left(\boldsymbol{i}_{\text{local}}^{(k,c)^\top} \cdot \boldsymbol{t}_{\text{global}}^{(k)}\right)}{\sum_{j=1}^{|\mathcal{P}|} \mathbf{1}_{[j \neq k]} \exp\left(\boldsymbol{i}_{\text{local}}^{(k,c)^\top} \cdot \boldsymbol{t}_{\text{global}}^{(j)}\right)} \tag{4}$$

This contrastive-based metric emphasizes representations similar to its paired image $\boldsymbol{t}_{\text{global}}^{(k)}$ (bigger nominator) and different from other incorrect pairings $\boldsymbol{t}_{\text{global}}^{(j)}$ (smaller denominator) in aggregation. We use $\text{softmax}$ for normalization and aggregate local image representations $\boldsymbol{i}_{\text{local}}^{(k,c)}$ as:

$$\alpha^{(k,1)}, \alpha^{(k,2)}, \ldots, \alpha^{(k,C)} = \text{softmax}(s^{(k,1)}, s^{(k,2)}, \ldots, s^{(k,C)}) \tag{5}$$

$$\boldsymbol{i}^{(k)} = \sum_c \alpha^{(k,c)} \cdot \boldsymbol{i}_{\text{local}}^{(k,c)} \tag{6}$$

#### 3.3.2 KNOWLEDGE ENSEMBLE TRANSFER

After aggregation, the server model distills knowledge from the clients by minimizing the $\ell_2$ distance between the output of the server $f(i^{(k)}; \mathbf{w})$ and the selectively aggregated representation $\boldsymbol{i}^{(k)}$:

$$\mathbf{w} := \mathbf{w} - \alpha \cdot \frac{1}{|\mathcal{P}|} \sum_{k=1}^{|\mathcal{P}|} \nabla \left\| f(i^{(k)}; \mathbf{w}) - \boldsymbol{i}^{(k)} \right\|_2 \tag{7}$$

where $\alpha$ is the learning rate. Our global-local cross-modal contrastive aggregation strategy benefits model learning from two perspectives: 1) by contrasting *local* representations to *global* representations that serve as consensus, the server can filter out outlier representations that drift too far, to mitigate the adverse impact induced by local model drift; 2) by contrasting to the paired data from another modality, our strategy can pick out representations that align better with its paired data, which helps the server model learn better multimodal representations.

# 4 EXPERIMENTS

To evaluate the proposed FL framework in multimodal setting, we approximate a real-life scenario (similar to Figure 1), where clients provide both uni-modal (images, text) and multimodal data (images tagged with descriptions). These diverse data sources are private, and cannot be disclosed during knowledge communication with the central server. Client models can handle either uni-modal tasks (e.g., image classification, text classification) or cross-modal tasks (e.g., multimodal retrieval). The goal is to apply federated learning over these heterogeneous clients to train a larger modal that can handle multimodal tasks on the server (e.g., multimodal retrieval). Sec. 4.1 describes the datasets and models used in our experiments to mimic this clients/server setting. Sec. 4.2 explains the evaluations on comparing our CreamFL framework with state-of-the-art FL methods. Sec. 4.3 and Sec. 4.4 provide further ablation studies and qualitative analysis on the model drift problem.

## 4.1 EXPERIMENTAL SETUP

**Datasets** The server model is evaluated on cross-modal retrieval and visual question answering tasks. We randomly choose a subset of MS-COCO (Lin et al., 2014) with 50,000 image-text pairs as public dataset. Details about datasets and evaluation are deferred to Appendix A.2. We distribute Flicker30K (Plummer et al., 2015) to 15 multimodal clients, CIFAR100 (Krizhevsky et al., 2009) to 10 unimodal image clients, and AGNEWS (Zhang et al., 2015) to 10 unimodal text clients, using Dirichlet distribution ($\alpha$=0.1) for non-IID data partition (Hsu et al., 2019). We randomly choose 10 from the total 35 clients to participate in training in each communication round.

**Baselines** We compare CreamFL with state-of-the-art FL approaches including: 1) FedAvg (McMahan et al., 2017), 2) FedIoT (Zhao et al., 2022), 3) FedMD (Li & Wang, 2019), 4) FedET (Cho et al., 2022), and 5) FedGEMS (Cheng et al., 2021). To demonstrate the effectiveness of our global-local contrastive aggregation (GCA) and local contrastive regularization (LCR), we further compare CreamFL with its two variants, which exclude the contrastive components (GCA & LCR) and instead use vanilla representation ensemble. We termed it "reamFL" (*CreamFL* without '*C*'). With different aggregation strategies, we have two additional baselines: 6) reamFL+Avg, which uses the same settings as CreamFL but without LCR and GCA, and the number of local samples is used for weighting in representation aggregation, same as FedAvg; 7) reamFL+IoT, which is similar to the first variant except that higher weights (set to 100 following Zhao et al. (2022)) are assigned to multimodal clients in representation aggregation, same as FedIoT.

All uni-modal algorithms are extended to multimodal scenarios by operating on each modality separately, *e.g.*, FedGEMS performs per-modality local distillation and entropy-based selective aggregation. For FedAvg and FedIoT, we distribute public data to multimodal clients and adopt client model of the same size as our server model for evaluation. For FedMD, we add the same large server model as CreamFL for comparing the server performance, keeping its averaging aggregation and local distillation unchanged. We choose ResNet-101 (He et al., 2016) and ResNet-18 as the server and client image models, respectively, and BERT (base) (Devlin et al., 2018) and GRU (Chung et al., 2014) as the text models. The representation dimension $d$ is 512 for both image and text. We use AdamP optimizer with initial learning rate 0.0002 and cosine learning rate scheduler for server model.

## 4.2 MAIN RESULTS

Table 1 presents Recall@1, Recall@5 and Recall@10 scores of all baselines and our method in multimodal retrieval task on 1K and 5K test sets of MS-COCO. As shown, our CreamFL framework achieves noticeable performance improvement over all baselines in all settings (on both image-to-text retrieval (i2t) and text-to-image retrieval (t2i) tasks).

Table 1: Comparison of CreamFL with baselines on image-text retrieval task.

| Types | Methods | 1K Test Images | | | | | | |
|---|---|---|---|---|---|---|---|---|
| | | i2t_R@1 | i2t_R@5 | i2t_R@10 | t2i_R@1 | t2i_R@5 | t2i_R@10 | R@1_sum |
| Model Homogeneous | FedAvg | 45.23 | 76.74 | 85.59 | 34.69 | 71.68 | 85.40 | 114.03 |
| | FedIoT | 43.31 | 75.62 | 86.26 | 33.94 | 70.09 | 84.56 | 111.19 |
| Model Heterogeneous (w/ larger server model) | FedMD | 48.40 | 80.24 | 89.64 | 38.23 | 74.44 | 86.68 | 128.33 |
| | FedET | 48.76 | 80.39 | 89.73 | 38.39 | 74.68 | 86.76 | 129.11 |
| | FedGEMS | 48.70 | 80.48 | 89.62 | 38.71 | 74.75 | 87.01 | 129.70 |
| | reamFL+Avg | 48.85 | 80.55 | 89.93 | 38.13 | 74.89 | 86.79 | 130.02 |
| | reamFL+IoT | 49.13 | 80.61 | 89.69 | 38.45 | 74.83 | 86.74 | 130.41 |
| | CreamFL (ours) | **49.66** | **80.66** | **90.13** | **38.94** | **75.02** | **87.14** | **132.88** |

| Types | Methods | 5K Test Images | | | | | |
|---|---|---|---|---|---|---|---|
| | | i2t_R@1 | i2t_R@5 | i2t_R@10 | t2i_R@1 | t2i_R@5 | t2i_R@10 |
| Model Homogeneous | FedAvg | 20.73 | 51.53 | 63.96 | 13.38 | 37.88 | 58.29 |
| | FedIoT | 20.64 | 50.81 | 63.45 | 13.30 | 38.49 | 56.23 |
| Model Heterogeneous (w/ larger server model) | FedMD | 23.95 | 53.23 | 66.29 | 17.73 | 44.46 | 58.47 |
| | FedET | 24.08 | 53.42 | 66.31 | 17.88 | 44.59 | 58.65 |
| | FedGEMS | 24.27 | 53.56 | 66.06 | 18.09 | 44.58 | 58.05 |
| | reamFL+Avg | 24.79 | 53.53 | 66.90 | 18.25 | 44.46 | 58.26 |
| | reamFL+IoT | 24.63 | 53.50 | 66.93 | 18.20 | 44.58 | 58.44 |
| | CreamFL (ours) | **25.34** | **53.62** | **66.95** | **18.94** | **44.68** | **58.82** |

In comparison to FedAvg and FedIoT, CreamFL exhibits superiority on both model performance (132.88% v.s. 114.03% & 111.19% on R@1_sum) and communication efficiency (Section 4.3), demonstrating the effectiveness and efficiency of the KD-based CreamFL framework. Compared with reamFL+IoT, which assigns higher weights to multimodal clients in aggregation, our method yields consistently better performance (132.88% v.s. 130.41% on R@1_sum score). This shows our global-local contrastive aggregation method for multimodal data is more effective. It is also more efficient as multimodal weights are computed automatically and no manual tuning is needed.

Comparing to FedMD, FedET and FedGEMS (algorithms with both aggregation and local regularization components), CreamFL also achieves superior performance (132.88% v.s. 128.33%, 129.11% & 129.70% on R@1_sum score). This demonstrates the effectiveness of our contrastive aggregation strategy and regularization method. The inferior performance of FedMD indicates that directly taking average of representations as ensemble, which may yield satisfying performance in uni-modal non-i.i.d. scenario, results in much worse performance in multimodal settings due to harsher model drift where simple average fails.

It is also worth noting that comparable performance of FedET and FedGEMS to FedAvg (KD) shows aggregation methods designed for logit with probabilistic properties (FedGEMS uses entropy and FedET uses variance of logit for weighting) may no longer have advantage in the context of aggregating representation, because representation does not share the same distribution nature as logit, revealing the limitation of logit-specific aggregation methods. We further validate the effectiveness

Table 2: VQA evaluation.

| Methods | Acc. |
|---|---|
| FedAvg | 52.54 |
| FedIoT | 53.06 |
| FedMD | 57.43 |
| FedET | 59.90 |
| FedGEMS | 60.23 |
| reamFL+Avg | 58.64 |
| reamFL+IoT | 59.64 |
| CreamFL (ours) | **62.12** |

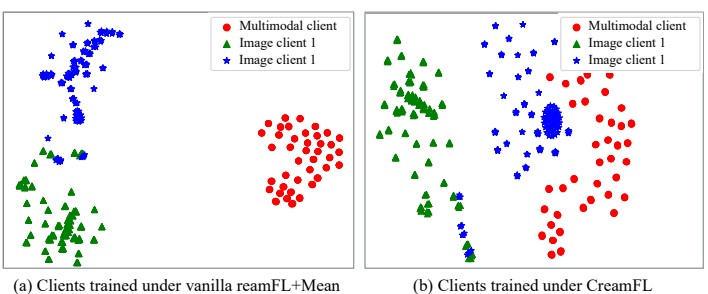

(a) Clients trained under vanilla reamFL+Mean  (b) Clients trained under CreamFL

Figure 3: T-SNE visualization (Van & Hinton, 2008) on model drift.

of CreamFL on VQA task, and present the results in Table 2. CreamFL yields an accuracy gain by 1.89% compared to the best baseline (62.12% v.s. 60.23%).

### 4.3 Ablation Studies

Table 3 studies the effect of each CreamFL component. *reamFL+Mean* adopts vanilla average as the representation aggregation strategy (all other settings the same as reamFL+Avg). *reamFL+GCA* replaces aggregation method with our strategy. *LCR.inter* and *LCR.intra* refer to two regularization techniques 3.2 for local client training. Results show reamFL+GCA not only surpasses vanilla average by a large margin (130.82% v.s.

Table 3: Results on ablation studies.

| Methods | R@1_sum |
|---|---|
| reamFL+Mean | 127.65 |
| reamFL+Avg | 130.02 |
| reamFL+IoT | 130.41 |
| reamFL+GCA | 130.82 |
| reamFL+GCA + LCR.inter | 131.41 |
| reamFL+GCA + LCR.intra | 131.02 |
| CreamFL (reamFL+GCA+LCR) | **132.88** |

127.65%), but also outperforms all baselines in Table 1, showing the effectiveness of our aggregation strategy. When local training regularization is added, the global performance can be further improved. *LCR.inter* outperforms *LCR.intra* by 0.41% (131.41% v.s. 131.02%), demonstrating that complementing information for the absent modality for uni-modal clients is critical in heterogeneous multimodal setting. Combing inter/intra-modal contrasts to regularize local training yields further performance boost.

Figure 4 shows the relation between communication cost and model performance. Two factors affect the communication: number of public data (*num*) and dimension of representation (*dim*). We vary each separately to control the communication. Specifically, for the blue lines, we fix *dim* as 512 and vary *num*, and vice versa for the orange line where *num* is fixed to 50k. X-axis is the communication cost in each round, with *dim* labeled on the upper axis and *num* on the lower. Each data point is the final model performance under the specified setting. Generally, FedAvg and FedIoT exhibit inferior performance to CreamFL with higher communication cost, and the amount of public data

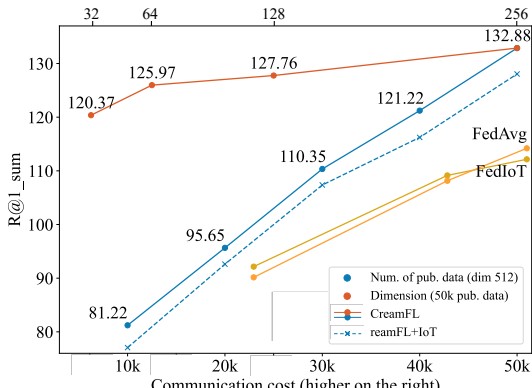

Figure 4: Communication v.s. performance.

(blue solid line) has a greater impact on performance than representation dimension (orange solid line). Other baselines also transmit representations on public data and have the same communication cost with CreamFL. Complete analysis is deferred to Appendix A.5.

### 4.4 Qualitative Study on Model Drift

Figure 3 visualizes representations of 250 randomly chosen images from COCO, generated by 1 multimodal client (red circle) and 2 different image clients (blue star and green triangle). (a) is a vanilla case where aggregation is simple average and no regularization is exerted on local training. We first observe that model drift exists between two modality-identical image clients (blue and green), in line with the observation in Karimireddy et al. (2020). Besides, model drift between uni-modal clients (blue v.s. green) is much smaller compared to the gap between multimodal and uni-modal clients (red v.s. blue+green), confirming our claim that modality gap and task gap cause further model drift problem in multimodal FL. (b) shows CreamFL pulls the representations generated by different clients closer and effectively mitigates such drift.

## 5 Conclusion

In this paper, we investigate multimodal federated learning and propose a new framework CreamFL that enables training larger server models and allows heterogeneous clients, via communicating knowledge on public dataset without disclosing private data and models. We transmit representations between server and clients and design a novel global-local cross-modal ensemble strategy with inter- and intra-modal contrastive regularization in local training to mitigate model drift.

ACKNOWLEDGEMENT

This work was supported by the National Key R&D Program of China under Grant No.2022ZD0160504, Xiaomi AI Innovation Research under grant No.202-422-002. We would also like to thank anonymous reviewers for their insightful comments.

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

# A   APPENDIX

## A.1   REGARDING THE USE OF PUBLIC DATA

Legal and copyright issues about the public data should be carefully taken into account in real-world applications. Obtaining permission is necessary before using public released data for commercial purposes. In general cases, the public data is chosen from publicly released datasets that are widely accessible. Clients normally acknowledge which public dataset the server uses, and can directly download them from public sources. For companies to apply this system, collecting their own public data is also possible. Regarding compliance and users' awareness, real-world applications (such as Google Assistant) often use FL on end clients who are opt-in [1]. Our KD-based FL framework can be implemented in a similar manner.

## A.2   DATASETS

A random subset of COCO (Lin et al., 2014) with 50,000 image-text pairs is selected as the public multimodal data. CIFAR100, AG_NEWS, and Flicker30k are used as the private datasets for image, text and multimodal clients, respectively. CIFAR-100 (Krizhevsky et al., 2009) consists of 50,000 colored training images in 100 classes, with 500 images each class. AG_NEWS (Zhang et al., 2015) contains 120,000 training sentences from 4 classes. Flicker30k (Plummer et al., 2015) contains 31,000 images collected from Flicker, together with 5 captions per image, *i.e.* 155,000 image-text pairs in total. For cross-modal retrieval, we follow Karpathy & Fei-Fei (2015) and report Recall@K results on 5K/1K test sets of MS-COCO, which measures the percentage of times a correct item being found among top-K results. For visual question answering, we use VQA v2.0 dataset (Goyal et al., 2017) and report accuracy over the 3,000 most frequent answers.

## A.3   DISCUSSIONS ABOUT REPRESENTATION ENSEMBLE DISTILLATION

Representation ensemble distillation has been an under-explored problem. Park & Kwak (2020) looked into this problem and their student network is trained to mimic the representations of all teachers after a non-linear transformation layer, but there is no active selection of teachers. This is implausible in the context of FL, because different clients (teachers) are highly heterogeneous in data distribution, and the quality of representations varies a lot (even malicious representations may exist). Thus, selectively aggregating these representations before performing representation-level distillation is required, and that is why we propose an aggregation strategy to address this challenge.

We believe that representation aggregation / ensemble is an important and unsolved problem in FL, especially when the community is embracing big foundation models (Devlin et al., 2018; Bommasani et al., 2021), where the server model in FL will need to learn foundational universal **representations** from heterogeneous data sources and diverse model architectures.

## A.4   LARGER DATA CAPACITY

We increase the number of public image-text pairs to 100,000 (also a random subset of MSCOCO), to evaluate under larger data capacity. Results in Table 4 show that CreamFL remains effective in the larger-scale setup. It is worth noting that the communication cost will increase linearly proportional to the public data number. Thus, improving model performance under the setting with less amounts of public data is an important future direction.

## A.5   TRADE-OFF BETWEEN COMMUNICATION AND PERFORMANCE

Trade-off results between communication and performance *w.r.t.* the number of public data samples for reamFL+IoT (the KD version of FedIoT), plotted as the blue dashed line in Figure 4. To further investigate the trade-off for other FL settings which transfer model parameters (such as FedAvg and FedIoT), we vary the model architecture of server model from ResNet101 to ResNet50 and ResNet34. Results show that CreamFL (orange and blue solid lines) exhibits a better trade-off compared with these frameworks.

---

[1]https://www.xda-developers.com/google-federated-learning-hey-google-accuracy/

Table 4: Results under larger data capacity.

| Methods | R@1_sum |
|---------|---------|
| FedGEMS | 151.85 |
| FedET | 151.68 |
| reamFL+Avg | 151.83 |
| reamFL+IoT | 151.57 |
| CreamFL | **152.50** |

## A.6 VISUALIZATIONS ON MODEL DRIFT

Figure 5 visualizes representations of 250 randomly chosen captions from COCO, generated by 1 multimodal client (red) and 2 different text clients (blue and green). Figure (a) is a vanilla case where clients are trained under reamFL+Mean and clients of (b) are trained under CreamFL. Similar to the observation in Figure 3 on image representations, we observe that model drift exists between two modality-identical text clients (blue and green), while this drift is much smaller than the gap between multimodal and uni-modal clients (red v.s. blue+green), confirming our claim that modality gap and task gap cause more severe model drift problem in multimodal FL. CreamFL effectively mitigate this drift as shown in (b).

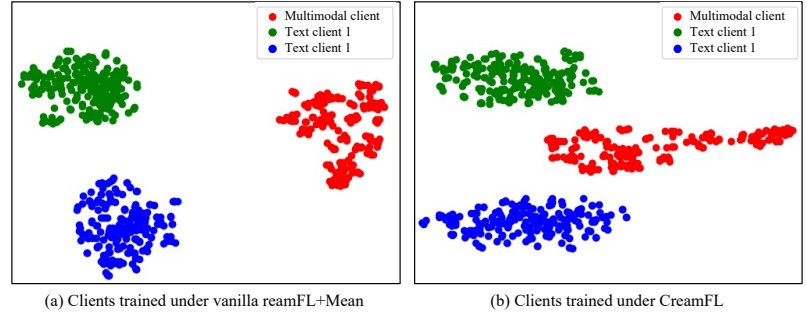

(a) Clients trained under vanilla reamFL+Mean      (b) Clients trained under CreamFL

Figure 5: T-SNE visualization of text representations on model drift.

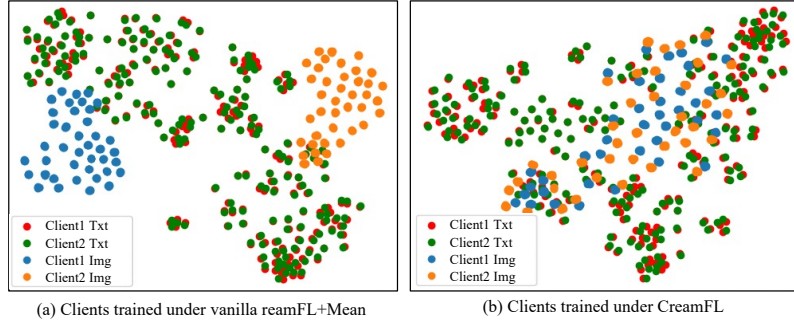

(a) Clients trained under vanilla reamFL+Mean      (b) Clients trained under CreamFL

Figure 6: T-SNE visualization of multimodal representations on model drift.

Further, representations of 250 randomly chosen image-text pairs from COCO are visualized in Figure 6, generated by two different multimodal clients. In Figure 6(a), clients are trained under reamFL+Mean and obvious drift on image representations can be observed (blue points are far away from orange ones). CreamFL effectively mitigate this drift as shown in (b) (blue and orange points are pulled together). Both frameworks exhibit less model drift on text representations of multimodal clients (red and green points).

## A.7 RELATED WORK

Federated Learning McMahan et al. (2017) is a privacy-conscious alternative to centralized ML, allowing multiple clients to collaboratively train models without compromising privacy. Existing FL work mainly focuses on: 1) improving server performance under complex non-iid scenarios (Li et al., 2020; Zhao et al., 2018; Karimireddy et al., 2020); 2) reducing communication cost from limited wireless bandwidth (Lin et al., 2017; Mao et al., 2022); and 3) protecting privacy (Melis et al., 2019; Zhu et al., 2019; Geyer et al., 2017) and enhancing server robustness against malicious clients (Yin et al., 2018; Bagdasaryan et al., 2020; Blanchard et al., 2017). Numerous aspects need to be taken into account to build a practical FL system, our work focus primarily on tackling the heterogeneity in multi-modal scenarios.

