# OpenReview forum: "Multimodal Federated Learning via Contrastive Representation Ensemble"
_ICLR.cc/2023/Conference — ICLR 2023 poster_

### Official Review · Reviewer_U8Yh · 2022-10-20

**Confidence:** 4
**Correctness:** 3
**Technical Novelty And Significance:** 4
**Empirical Novelty And Significance:** 3
**Recommendation:** 6

**Clarity, Quality, Novelty And Reproducibility:**

Clarity & Quality: Quality of the results are adequate however, a few details can be clarified as highlighted in the weaknesses

Novelty: The proposed approach is novel enough.

Reproducibility: The presentation and the details provided are reasonable to reproduce however having a bit more clarity will improve the reproducibility.

**Strength And Weaknesses:**

# Pros:

- The paper is well-written, easy to understand and follow along.

- The more interesting aspect is the way the modality and task gaps are treated through regularized losses and the ensemble strategy, that is the global-local and cross-modality contrastive aggregation.

- The aggregation in a heterogeneous setting, where some clients are uni-modal and some are multimodal is also complex and the proposed approach seem to deal with such complex scenarios.

- Literature is covered adequately.

# Cons:

- Serious concern, use of public data is questionable here from a real world perspective. When the client data is private and public data has licenses in place one has to make sure they comply since the clients are unaware of this public data usage at the server or even worse some of the data seem to get transferred to clients as well, if not mistaken from the explanation in section 4.

- The proposed regularizations in the loss sounds more useful for a multimodal client rather than a unimodal client.

- If you already know a given client is image-modal then you can make use of the global image representations and train on them and communicate only them back, what is the point of using text-modal representations in that case at all. Similarly for clients with text only modality?

- In other words, what is the point of l_{inter} how does pushing the uni-model representations towards multi-modal representations help close the multi-modal gap? This obviously going to degrade the performance on the image tasks (text in text case).

- What is this with and without larger server model in Table 1, tried to find it else where in the text the difference can not identify, needs clarity on this.

- The theme is to improve performance on multimodal test sets, with reamFL+IoT, where you increase the multimodal client weightage, the performance is surprisingly worse than the reamFL+Avg, any reasons?

- At any point of time, be it the server or the client, the technique seem to rely on storing two or more copies of the global and client models, which warrants access to more resources which opposes the motivation in the Introduction that clients are often resource constrained.

- This approach is to train representations in FL in multimodal context, then the details about how the fine-tuning is done and how long it is done and what is the data used for fine-tuning on the down-stream tasks is missing.

- The real ablation that will be interesting to see is how does CreamFL perform when you decrease the uni-modal clients, further you can permute with the number of image and text clients when you decrease the total number of uni-modal clients. Then, what if you mute the l_{inter} on the corresponding uni-modal clients but still use the creamFL framework?


- How is the ensemble strategy different from the FedAvg style except that you use L2 for distillation?, in which case, why should this be titled or called an ensemble?


**Summary Of The Paper:**

The paper proposes Contrastive Representation Ensemble and Aggregation for Multimodal
FL (CreamFL) to exploit multimodal data from clients in FL settings in a privacy driven world. CreamFL trains large models from clients with heterogeneous architectures and multiple modalities. To fuse multimodal representations, a global-local cross-model ensemble strategy is proposed.


**Summary Of The Review:**

Overall, the paper is a decent attempt to address a complex problem however, given the concerns highlighted in the strengths and weaknesses, would like to weight for the response from the authors before changing my scores.

---

> ### Author Response · Authors · 2022-11-17
> **Response to reviewer U8Yh (Part 3/3)**
>
> **Q7: Details about how time cost and datasets of fine-tuning.**
>
> Although the server model learns representations from client models, we need to clarify that we do not follow the pretrain-finetune paradigm. Instead, each task is trained in a traditional way in each communication round. The server first trains on its own datasets (line 11 of Algorithm 1), then leverages public datasets (e.g., COCO) to distill knowledge from clients (line 13 of Algorithm 1). In our experiments, COCO and VQA v2.0 are used for retrieval and VQA tasks, respectively, as detailed in Section 4.1. CreamFL costs about 102 minutes for a full communication round that contains sequential global and local training on a single NVIDIA-A30 GPU.
>
> Incorporating pretraining and big foundation models is a promising future direction for FL, and our work serves as a stepping stone towards that ultimate goal. Since more and more data are generated from mobile and edge devices with privacy and computation constraints, we do believe federated aggregation of representations will be a critical technique worth studying to further empower large foundation models.
>
> **Q8: How is the ensemble strategy different from the FedAvg style except that you use L2 for distillation? Why should this be titled or called an ensemble?**
>
> The ensemble strategy of FedAvg-style algorithms is to use the number of local data to weigh different clients in the aggregation. To mitigate the local drift problem, we propose a novel global-local cross-modal *contrastive* strategy that automatically computes *sample-wise* contrastive scores to weigh each sample from different clients during ensemble, which is more effective than all existing ensemble strategies. More details can be found in Section 3.3 in our paper.
>
> Note that ensemble is irrelevant to L2 distillation. The server action of KD-based Federated Learning system consists of two steps: $1)$ the server receives representations from all clients and then *selectively ensembles* these representations, because the quality of representations across clients varies; $2)$ the server distills knowledge from the ensembled representations through L2 loss. The first phase is commonly termed as 'aggregation' [14] or 'ensemble' [2] in literature.
>
> **References**
>
> [1] Daliang Li and Junpu Wang. Fedmd: Heterogenous federated learning via model distillation. arXiv preprint arXiv:1910.03581, 2019.
>
> [2] Yae Jee Cho, et al. Heterogeneous ensemble knowledge transfer for training large models in federated learning. IJCAI 2022.
>
> [3] Tao Lin, et al. Ensemble distillation for robust model fusion in federated learning. NeurIPS 2020.
>
> [4] Xuan Gong, et al. Ensemble attention distillation for privacy-preserving federated learning. ICCV 2021.
>
> [5] Alysa Ziying Tan, et al. Towards personalized federated learning. IEEE Transactions on Neural Networks and Learning Systems, 2022.
>
> [6] Hong-You Chen, et al. On bridging generic and personalized federated learning for image classification. ICLR, 2022.
>
> [7] Yen-Chun Chen, et al. Uniter: Universal image-text representation learning. ECCV 2020.
>
> [8] Xiujun Li, et al. Oscar: Object-semantics aligned pre-training for vision-language tasks. ECCV 2020.
>
> [9] Wonjae Kim, et al. Vilt: Vision-and-language transformer without convolution or region supervision. ICML 2021.
>
> [10] Zirui Wang, et al. Simvlm: Simple visual language model pretraining with weak supervision. ICLR 2022.
>
> [11] Junnan Li, et al. Align before fuse: Vision and language representation learning with momentum distillation. NeurIPS 2021.
>
> [12] Amanpreet Singh, et al. Flava: A foundational language and vision alignment model. CVPR 2022.
>
> [13] Wenhui Wang, et al. Image as a foreign language: Beit pretraining for all vision and vision-language tasks. arXiv preprint arXiv:2208.10442, 2022.
>
> [14] Hongyan Chang, et al. Cronus: Robust and heterogeneous collaborative learning with black-box knowledge transfer. arXiv preprint arXiv:1912.11279, 2019.

---

> ### Author Response · Authors · 2022-11-17
> **Response to reviewer U8Yh (Part 2/3)**
>
> **Q3: The regularization sounds more useful for multimodal clients than uni-modal clients. Experiments on decreasing the number of uni-modal clients and muting $l_{inter}$ for uni-modal clients in CreamFL.**
>
> We vary the ratio of uni-modal and multimodal clients to analyse the effect of composition of clients. The results are presented in the table below. The original setting of CreamFL (4th row) contains 10 image clients, 10 text clients, and 15 multimodal clients. We first halve the number of uni-modal and multimodal clients separately (1st and 2nd row), then reduce them simultaneously (3rd row). As a result, the total number of private data samples decreases accordingly. From the table below, we can draw the following conclusions: firstly, decreasing the number of clients leads to performance deterioration; secondly, reducing the number of multimodal clients results in a sharper decline in model performance as compared to reducing uni-modal clients, indicating that multimodal clients may contribute more to the final federated model.
>
> |  |   Number of clients    |                    | R@1\_sum  |
> | :---------------: | :---------: | :----------------: | :-------: |
> |    Img clients    | Txt clients | Multimodal clients |           |
> |         5         |      5      |         15         |   91.94   |
> |        10         |     10      |         7          |   90.83   |
> |         5         |      5      |         7          |   90.24   |
> |        10         |     10      |         15         | **92.43** |
>
> In an additional experiment, we mute $l_{inter}$ for uni-modal clients to test the effectiveness of inter-modal contrast on uni-modal clients. Besides conducting experiments under the original settings of our paper, we further evaluate under a larger-scale setting where the text and image encoders of the server are replaced by BERT and ResNet101.
>
> | Methods                             | R@1_sum |
> | :---------------------------------- | :-----: |
> | CreamFL                             |  92.43  |
> | CreamFL $-\ l_{inter}^{uni}$        |  91.11  |
> | CreamFL (BERT)                      | 132.36  |
> | CreamFL (BERT) $-\ l_{inter}^{uni}$ | 130.58  |
>
> Results show that muting $l_{inter}$ for uni-modal clients consistently brings obvious performance deterioration, suggesting that inter-modal contrasts on uni-modal clients play an important role in mitigating the modality gap. The code of this ablation study will be released together with main experiments to facilitate further study.
>
> **Q4: What's with and without larger server model in Table 1?**
>
> 'With larger server model' means the algorithms can  accommodate heterogeneous client-side model architectures and can train a server model larger than client-side models. 'Without larger server model' means these algorithms require homogeneous model architecture across server and clients because model parameters are transferred across server/clients, such as in FedAvg. We have revised Table 1 to clarify the differences in the updated version.
>
> **Q5: The theme is to improve performance on multimodal test sets, why reamFL+IoT that increases the multimodal weights performs worse than reamFL+Avg?**
>
> The inferiority of reamFL+IoT is due to its sensitivity to the multimodal weight. In our implementation, we followed the original FedIoT paper to set the multimodal weight as 100 without hyper-parameter tuning. Additional results with comprehensive tuning are presented in the table below. Numbers better than reamFL+Avg (88.13) are in bold. Compared with this method, our aggregation weights are automatically computed and more effective.
>
> | Multimodal weight |    10     |    20     |  25   |  50   |    75     |    90     |  100  |  125  |    150    |  175  |  200  |  500  |
> | :---------------: | :-------: | :-------: | :---: | :---: | :-------: | :-------: | :---: | :---: | :-------: | :---: | :---: | :---: |
> |      R@1_sum      | **88.29** | **88.31** | 87.09 | 88.07 | **88.21** | **88.35** | 88.05 | 87.01 | **88.27** | 86.87 | 87.00 | 87.53 |
>
> **Q6: Storing two or more copies of the global and client models opposes the motivation that clients are resource constrained.**
>
> We do not store two or more copies of the global and client models. Each local client only stores its local model. To perform intra-modal contrast, each client needs to store low-dimensional representations of public data. Similarly, the server only stores the server-side model. Note that the motivated spirit of our heterogeneous CreamFL is to support arbitrary client model design, so that client model architecture and the quantity of public data can be flexibly customized based on local resource constraints.

---

> ### Author Response · Authors · 2022-11-17
> **Response to reviewer U8Yh (Part 1/3)**
>
> Thanks for your constructive comments. Our detailed response can be found below.
>
> **Q1: Serious concern, use of public data is questionable from a real world perspective. When the client data is private and public data has licenses in place one has to make sure they comply since the clients are unaware of this public data usage at the server or even worse some of the data seem to get transferred to clients as well, if not mistaken from the explanation in section 4.**
>
> Thanks for your comments. We would like to clarify that in general cases of FL the public data is often chosen from publicly released datasets which are widely accessible. Clients normally acknowledge which public dataset the server uses, and the server does not have to transfer *any public data* to clients as they can directly download them from public sources. What is being transferred in FL is the clients' knowledge on this public data. Regarding compliance and users' awareness, real-world applications (such as Google Assistant) often use FL on end clients who are opt-in [(url)](https://www.xda-developers.com/google-federated-learning-hey-google-accuracy/). Our KD-based FL framework can be implemented in a similar manner. This is a well-defined problem [1] and there has been a line of studies on KD-based FL leveraging freely available public data for transferring knowledge. The background and setting of this line of research has been widely accepted by the community [1,2,3,4].
>
> **Q2: How does the inter-modal contrast help close the multimodal gap? This is obviously going to degrade the performance on the uni-modal tasks.**
>
> Thank you for the question.
>
> Uni-modal clients trained under a single modality have never seen data from another modality in local training. This calls for a necessary compensation for the information of absent modalities. Inter-modal regularization contrasts local representations with the aligned representations from another modality, thus introducing additional information from other modalities into local training, serving this compensation purpose. This intuition is detailed in Section 3.2.1. Its effectiveness is validated through main experiments in Section 4 in the paper, and by additional experiments on muting the inter-modal contrast for uni-modal clients (more details can be found in the following Q3 response).
>
> **Degrade the performance on the client image tasks:**
>
> This is indeed a real concern, while we humbly remind that our goal is to collaboratively train a larger server model for multimodal tasks.
>
> 1. We aim at training a strong global server model and did not focus on clients' local performance. The FL community is divided into two camps by this training goal [5,6]: 1) Generic FL that aims at training a global server model; 2) Personalized FL that pursues better performance on client models. CreamFL falls into the first category. It is commonly believed that a trade-off exists between the performances of global and local models,  because each client alone is highly non-iid and biased [6]. If a client excels at its specific task, it is natural that it lacks global generalization and may contribute poorly to the global model performance. It will be a valuable contribution to the community in and of itself to improve model performance on both server and client tasks. Given that our goal in this paper is to train a large multimodal server model, slightly sacrificing local uni-modal client performance on uni-modal tasks is tolerable in our setting.
>
> 2. We focus on improving on **multimodal** tasks. Building a system that excels indiscriminately at image tasks, text tasks and multimodal tasks is a challenging research problem. Most previous works [7,8,9,10,11] only target at multimodal tasks and did not concern about the performance on uni-modal tasks. Very recent studies [12,13] achieved promising performance on multimodal and one uni-modality task, but with degraded performance on another uni-modality.
>
> We sincerely acknowledge that this is an important problem to tackle, and we aspire to design a more powerful all-round FL system in future work, which will be a valuable contribution to both multimodal learning (convergence on vision, language, cross-modal tasks) and federated learning (convergence of global server and local clients performances).

---

> ### Author Response · Authors · 2022-11-30
> **Follow-up. Thanks for the questions again.**
>
> Thanks again for the detailed comments and suggestions, which have greatly helped us in improving the quality and clarity of our work. Additional experiments, clarifications about the scope of our paper, solutions to the concerns of real world applications have been added. We would really appreciate it if you could let us know whether your concerns have been addressed and if we can clarify anything else. If there are any further questions, we'd be happy to engage in further discussions and answer them.
>
> Thanks!

---

### Official Review · Reviewer_FnzN · 2022-10-22

**Confidence:** 5
**Correctness:** 3
**Technical Novelty And Significance:** 3
**Empirical Novelty And Significance:** 3
**Recommendation:** 8

**Clarity, Quality, Novelty And Reproducibility:**

The article has fluent language, clear writing logic, and certain innovation, while the author submitted the code for the implementation of the algorithm.

**Strength And Weaknesses:**

Strengths: The authors propose contrastive representation ensemble and aggregation for multimodal federated learning. The innovative points of the article are novel and practical, the writing is logical and clear with beautiful figures and adequate experiments.
Weaknesses:
(1)	This paper focuses on image-text multimodal learning, and the authors should fully describe the image-text multimodal learning algorithms and analyze the similarities and differences with the algorithms in this paper, such as the differences between the intra-modal contrast learning and inter-modal contrast learning in other algorithms and this paper, and the differences between other multimodal representation fusion algorithms and this paper.
(2)	The authors point out that existing multimodal FL methods limit the global model in terms of model complexity and data capacity, so it is desired that the authors increase the model complexity and data capacity analysis of the proposed method and compare it numerically with other algorithms.


**Summary Of The Paper:**

The paper proposes comparative representation integration and aggregation for multimodal FL (CreamFL). In particular, the global-local cross-modal contrastive aggregation strategy is used to achieve better multimodal representation fusion. Meanwhile, regularizing local training is employed to mitigate local model drift caused by modal differences and task differences. CreamFL is the first multimodal FL framework based on knowledge distillation that supports heterogeneous schemas and model architectures between servers and clients, while exchanging private knowledge only on public datasets, without leaking private models and data. And CreamFL can train larger models on the server and absorb modality-diverse knowledge from resource-constrained clients.

**Summary Of The Review:**

The article proposes a comparative representation integration and aggregation of multimodal FL based on Knowledge Distillation, which supports heterogeneous schemas and model architectures between servers and clients and can absorb model-diverse knowledge from resource-constrained clients. And the proposed approach allows the exchange of private knowledge on public datasets without disclosing private models and data. The innovative points of the article are novel and practical, the writing is logical and clear, and the experiments are adequate and promising. To further improve the quality of this manuscript, it is hoped that the authors can add some theoretical analysis and comparisons with other image-text multimodal learning algorithms.

---

> ### Author Response · Authors · 2022-11-17
> **Response to reviewer FnzN**
>
> Thank you for the supportive review. Below we address the detailed comments.
>
> **Q1: Analyse similarities and differences between algorithms in this paper and other image-text multimodal learning algorithms, such as the intra-modal contrast, inter-modal contrast and multimodal representation aggregations.**
>
> Inter-modal contrastive learning is a prevailing technique for multimodal self-supervised learning [1,2,3,4]. The contrastive part of most pre-training frameworks resembles that of CLIP [2], where aligned and unaligned image-text pairs are treated as positive and negative pairs, respectively. Our method differs from them in two main aspects: 1) we creatively apply the concept of contrasting to design a ''metric'' for evaluating the quality of representations, to address the challenge that the quality of representations varies significantly between clients due to their inevitable heterogeneity; 2) we propose to use inter-modal contrast as a ''regularization'' technique to restrict global-local discrepancy, which also bridges the modality gap between clients.
>
> Few multimodal algorithms use intra-modal contrast. Different from traditional uni-modal contrastive learning [5,6] that minimizes the representations of two views of the same sample, our intra-modal contrastive algorithm minimizes the representations of the same input generated by different ``models'' (between global server and local clients) to address model drift.
>
> Representation aggregation distillation has also been under-explored. [7] looked into this problem and their student network is trained to mimic the representations of all teachers after a non-linear transformation layer, but there is no active selection of teachers. This is implausible in the context of FL, because different clients (teachers) are highly heterogeneous in data distribution, and the quality of representations varies a lot (even malicious representations may exist). Thus, selectively aggregating these representations before performing representation-level distillation is required, and that is why we propose an aggregation strategy to address this challenge. To the best of our knowledge, there are few previous studies considering multimodal representation aggregation.
>
> We believe that representation aggregation is an important and unsolved problem in FL, especially when the community is embracing big foundation models, where the server model in FL will need to learn foundational universal representations from heterogeneous data sources and diverse model architectures.
>
> **Q2: Experiments and analysis of larger model complexity and data capacity.**
>
> We replace GRU and ResNet50 with BERT and ResNet101 as the text and image server models, respectively, to evaluate under larger model complexity. The computational resources of clients are generally limited in real scenarios, so we keep the current setting of local models. Results show that CreamFL remains effective in the larger-scale setup.
>
> | Methods    | R@1_sum |
> | :--------- | :-----: |
> | FedGEMS    | 128.33  |
> | FedET      | 129.11  |
> | FedMD      | 129.70  |
> | reamFL+Avg | 130.02  |
> | reamFL+IoT | 130.41  |
> | CreamFL    | **132.88**  |
>
> Additional experiment under larger data capacity takes more time. Please allow us to include it in final version. Theoretical analysis of contrastive techniques under the framework of Federated Learning has been under-explored in the literature. We will investigate it with deeper probing into multimodal federated learning in future work. Thanks for your suggestions!
>
> **References**
>
> [1] Chao Jia, et al. Scaling up visual and vision-language representation learning with noisy text supervision. ICML 2021.
>
> [2] Alec Radford, et al. Learning transferable visual models from natural language supervision. ICML 2021.
>
> [3] Amanpreet Singh, et al. Flava: A foundational language and vision alignment model. CVPR 2022.
>
> [4] Wenhui Wang, et al. Vlmo: Unified vision-language pre-training with mixture-of-modality-experts. NeurIPS 2022.
>
> [5] Kaiming He, et al. Momentum contrast for unsupervised visual representation learning. CVPR, 2020.
>
> [6] Ting Chen, et al. A simple framework for contrastive learning of visual representations. ICML 2020.
>
> [7] SeongUk Park, et al. Feature-level ensemble knowledge distillation for aggregating knowledge from multiple networks. ECAI 2020.

---

> ### Author Response · Authors · 2022-11-30
> **Follow-up. Thanks for your suggestions again.**
>
> Thanks for the positive feedback again. The discussion about other multimodal learning algorithms and experiments under larger data capacity will be included in the final version. If there are any future questions, we’d be happy to answer them.
>
> Thanks!

---

### Official Review · Reviewer_KYph · 2022-10-25

**Confidence:** 4
**Correctness:** 3
**Technical Novelty And Significance:** 3
**Empirical Novelty And Significance:** Not applicable
**Recommendation:** 5

**Clarity, Quality, Novelty And Reproducibility:**

The structure of the article is clear and easy to understand. The experimental design answers the main considerations of the proposed method.
This article has made improvements and innovations in the methods of the past literature, and applied it to the challenging multimodal federated learning, showing a certain degree of innovation. However, the core techniques are not novel, though I understand the application scenario is interesting.
The author has uploaded their source code on the supplementary materials, which can help others reproduce the work.


**Strength And Weaknesses:**

Strength:
1. This paper provides insight into the new heterogeneity challenges caused by modality heterogeneity, which is unique for multi-modal federated learning. The authors emphasize on modality gap and task gap between uni-modal clients and multi-modal clients.
2. By introducing a public dataset in the server, the proposed method further extends the idea of contrastive representation among the local representations to the contrast between the global representation and local representation. The contrastive information is used for local training and global aggregation.
3. The proposed knowledge ensemble transfer schema utilizes the aggregated local representation to update the global model parameters. This overcomes the challenge that the client owns different model architectures.
Weakness:
1. Incomplete experiment design. For the analysis of communication cost and model performance part, the communication cost of both model parameters and transmission of the public dataset for baseline models is expected to be given. As the local models and global model changes, the tradeoff between communication cost and performance is underexplored.
2. From my understanding, the main technique part is based on KD and ideas from contrastive learning. I got the point that the statement saying these techniques are applied in multimodal FL is interesting. However, the novelty is still the most concerning for me.
3. For the analysis of model drift, the authors only illustrate the modality gap for the image representation. The analysis of text representation and multi-modal representation is expected.
4. Lack of analysis of how the composition of the clients will affect the global model performance.
5. The introduction of the public dataset will endanger the system's security and privacy protection.


**Summary Of The Paper:**

This paper proposed a knowledge-distillation-based federated learning framework, CreamFL, for a multimodal setting where clients may hold different combinations of data modalities. The inter-modal and intra-modal contrast losses are used as a regularization term to mitigate the model drifting issue, which is called the modality gap and task gap. The CreamFL utilizes the representations of a public dataset to enhance the aggregation strategy and achieves a state-of-art performance in the multi-modal setting with heterogeneous modality and model architectures.


**Summary Of The Review:**

This paper connects clients and servers with different target tasks through a public dataset, thereby enabling knowledge distillation to train a larger global model from each client. The extended contrast idea utilizes the global representation and local representation to overcome the novel modality heterogeneity challenge. While there is a lack of well-established experiment design to demonstrate the superiority of the model on communication cost tradeoff and privacy protection. Also, I am concerned about the technique's novelty as well. I would not think it reaches the acceptance bar of this conference.

---

> ### Author Response · Authors · 2022-11-17
> **Response to reviewer KYph (Part 2/2)**
>
> **Q5: Public datasets endanger the system's privacy and security.**
>
> We argue that exchanging knowledge about public datasets is, on the contrary, safer than the typical line of federated learning algorithms that transmits model parameters (e.g., FedAvg).
>
> FedAvg-style framework completely exposes clients' model parameters to others. The shared parameters leak a significant amount of information about the local private data [1,2], and in certain cases pixel/token-wise private data can be recovered through gradient/model inversion attacks [3,4,5,6]. In contrast, transmitting knowledge through an external public dataset treats local models as black-boxes and does not explicitly share their model architecture or private data, largely protecting the privacy of local clients. In addition, FedAvg-style framework is vulnerable to backdoor attacks [7], where some participants can maliciously influence the aggregated model parameters directly. Many existing works show that KD-based FL is more robust to data inference [8,9] and backdoor attacks [9], in agreement with our observation.
>
> **References**
>
> [1] Milad Nasr, et al. Comprehensive privacy analysis of deep learning: Passive and active white-box inference attacks against centralized and federated learning. In 2019 IEEE symposium on security and privacy (SP), pp. 739–753. IEEE, 2019.
>
> [2] Luca Melis, et al. Exploiting unintended feature leakage in collaborative learning. In 2019 IEEE symposium on security and privacy (SP), pp. 691–706. IEEE, 2019.
>
> [3] Ligeng Zhu, et al. Deep leakage from gradients. NeurIPS 2019.
>
> [4] Bo Zhao, et al. idlg: Improved deep leakage from gradients. arXiv preprint arXiv:2001.02610, 2020.
>
> [5] Jonas Geiping, et al. Inverting gradients-how easy is it to break privacy in federated learning? NeurIPS 2020.
>
> [6] Hongxu Yin, et al. See through gradients: Image batch recovery via gradinversion. CVPR 2021.
>
> [7] Eugene Bagdasaryan, et al. How to backdoor federated learning. In International Conference on Artificial Intelligence and Statistics, pp. 2938–2948. PMLR, 2020.
>
> [8] Daliang Li and Junpu Wang. Fedmd: Heterogenous federated learning via model distillation. arXiv preprint arXiv:1910.03581, 2019.
>
> [9] Hongyan Chang, et al. Cronus: Robust and heterogeneous collaborative learning with black-box knowledge transfer. arXiv preprint arXiv:1912.11279, 2019.

---

> ### Author Response · Authors · 2022-11-17
> **Response to reviewer KYph (Part 1/2)**
>
> Thanks for spending the time on our submission and the valuable feedback. We hope our rebuttal can clarify your concerns.
>
> **Q1: Communication costs of both model parameters and public dataset for baseline models is expected. As the local models and global model changes, the tradeoff between communication and performance is underexplored.**
>
> Thanks for your comments. More ablation results are included in Section A.4 of Appendix.
>
> CreamFL and other KD-based FL baselines transfer knowledge through public data, and no model parameters are transmitted. Two factors that influence the communication cost in CreamFL have been investigated: representation dimension and the number of public data samples. We add the trade-off experiments *w.r.t.* the number of public data samples for reamFL+IoT (the KD version of FedIoT). Results show that CreamFL exhibits a better trade-off between model peformance and total communication bits. To further investigate the trade-off for other FL settings which transfer model parameters (such as FedAvg and FedIoT), we vary the model architecture of server model from ResNet50 to ResNet34 and ResNet18. Results show that CreamFL also achieves better trade-off compared with these frameworks. Complete results can be found in Appendix A.4.
>
> **Q2: Novelty not enough.**
>
> Although our framework is based on KD and contrastive learning, we are not directly implementing KD and contrastive learning. Our contributions include both novel methodologies and valuable insights, summarized as follows:
>
> Firstly, different from the traditional way that contrastive learning is used in unsupervised learning, we creatively design a cross-modal **metric** for dynamically evaluating the quality of representations from heterogeneous clients, for later aggregation. In addition, inter-modal contrast is proposed as a **regularization** technique to restrict the observed global-local discrepancy and bridge the modality gap between clients, which is empirically proved as more effective than existing FL methods.
>
> Secondly, different from how KD is traditionally applied to FL frameworks, our CreamFL communicates the representations of public data instead of logits, which allows for more general and diverse tasks such as multimodal retrieval, whereas traditional KD-based FL algorithms are mostly limited to classification tasks. In addition, CreamFL supports larger server model training and enjoys better privacy and security than FedAvg-type FL frameworks, as explained in Q5 below.
>
> Thirdly, we are the first to probe deeply into multimodal FL (MMFL) and to identify unprecedented challenges newly emerged in multimodal federated learning, such as modality gap and task gap, and propose practical and effective solutions to address these challenges. The MMFL setting considered in our paper is practical and non-trivial, especially as more and more diverse-formatted data with privacy constraints are generated from AIoT and mobile edge devices nowadays, while large models can only be trained on centralized and resource-abundant server side. Our work contributes to pushing the envelope of training larger models with small, fragmented, heterogeneous and private data.
>
> **Q3: Model drift analysis of text and multi-modal representation.**
>
> Thanks for your suggestions. We add analysis of the modality gap between uni-modal text clients and multimodal clients, as well as the analysis on model drift between different multimodal clients. Results are included in Section A.5 of Appendix.
>
> **Q4: How the composition of the clients affects the global model performance.**
>
> We vary the ratio of uni-modal and multimodal clients to analyse the effect of composition of clients. The results are presented in the table below. The original setting of CreamFL (4th row) contains 10 image clients, 10 text clients, and 15 multimodal clients. We first halve the number of uni-modal and multimodal clients separately (1st and 2nd row), then reduce them simultaneously (3rd row). As a result, the total number of private data samples decreases accordingly. From the table below, we can draw the following conclusions: firstly, decreasing the number of clients leads to performance deterioration; secondly, reducing the number of multimodal clients results in a sharper decline in model performance as compared to reducing uni-modal clients, indicating that multimodal clients may contribute more to the final federated model.
> |  |    Number of clients         |                    | R@1\_sum  |
> | :---------------: | :---------: | :----------------: | :-------: |
> |    Img clients    | Txt clients | Multimodal clients |           |
> |         5         |      5      |         15         |   91.94   |
> |        10         |     10      |         7          |   90.83   |
> |         5         |      5      |         7          |   90.24   |
> |        10         |     10      |         15         | **92.43** |

---

> ### Author Response · Authors · 2022-11-30
> **Follow-up. Thanks for your review again.**
>
> We thank the reviewer again for the helpful and detailed review. Additional experiments have been added, concerns about novelty and threat of public datasets to system's security and privacy have been clarified. We sincerely hope our corrections, clarifications, and additional results could address your concerns. If there are any further questions, please let us know, we'd be glad to address them.
>
> Thanks!

---

### Official Review · Reviewer_LDTA · 2022-10-26

**Confidence:** 4
**Correctness:** 4
**Technical Novelty And Significance:** 4
**Empirical Novelty And Significance:** 3
**Recommendation:** 6

**Clarity, Quality, Novelty And Reproducibility:**

The proposed method is based on the FedET and K-D based federated learning and the core of the method i.e., LCR and GCA, works fine and quite interesting for multimodal federated learning.

**Strength And Weaknesses:**

Weakness:

1. The main results are not quite enough. For instance, the discrepancy between public datasets and private datasets is not taken into account and hence more public/private datasets should be evaluated. What’s more, more practical and complex scenarios could also be tested, e.g., assigning different private datasets for different image/text/multi-modal clients.

2. What are the communication costs of FedMD, FedET and others? Are they more efficient than CreamFL?

3. The ablation results are not complete, e.g., the results of the reamFL+LCR are missing.

Strength:

1. The submission proposed a novel heterogeneous multimodal federated learning framework firstly and achieves a more promising result than prior works. Multi-modal federated learning is a subarea under exploration and this work firstly proposes an attempt to extend K-D distilled-based method to multimodal federated learning.

2. A novel feature aggregation strategy, namely GCA, and local-global contrastive objectives, namely LCR, are designed for ensemble knowledge distillation and addressing the model drift for multi-modal federated learning, which is quite interesting for multimodal federated learning.


**Summary Of The Paper:**

In this submission, the authors proposed a heterogeneous federated learning framework to handle the heterogeneous model architectures and data modalities for diverse training clients. Specifically, the proposed CreamFL has a novel cross-modal contrastive feature aggregation strategy and the inter-/intra-model contrastive objective to address the model drift for multi-modal federated learning. The experiments show the advances of the proposed method on efficiency and performance in multi-modal settings for federated learning.

**Summary Of The Review:**

Please see the Strength And Weaknesses.

---

> ### Author Response · Authors · 2022-11-17
> **Response to reviewer LDTA**
>
> Thank you for the comments! We have uploaded a revision of our paper. Below we address the detailed questions.
>
> **Q1: Studies about the discrepancy between public and private datasets, and more practical
> and complex scenarios that assign different private datasets for clients.**
>
> **Discrepancy between public and private datasets:**
>
> In the current experimental setting, a random subset of COCO is selected as the public multimodal data. CIFAR100, AG\_NEWS, and Flicker30k are used as the private datasets for image, text and multimodal clients, respectively.
> To study how the discrepancy between public and private data affects model performance, we replace Flicker30k with another subset of COCO containing 150k image-text pairs (similar size to Flicker30k that contains 155k image-text pairs) as the multimodal private data, which has high similarity to the public dataset COCO used on the server. We make sure that this subset of COCO has no overlap with the public data. Other settings such as the number of multimodal clients are unchanged. Results are presented on the left side of the table below ("COCO+COCO"). Our method consistently achieves superior performance under this setup. We further compare with the main experiments in our paper ("COCO+Flicker30K" in the table below). These results corroborate the assumption that smaller discrepancy between public and private data leads to better performance.
>
> | COCO+COCO  |  | COCO+Flicker30K |  |
> | :--------- | :-------: | :-------------- | :-------------: |
> | Methods    |  R@1_sum  | Methods         |     R@1_sum     |
> | FedGEMS    |   90.21   | FedGEMS         |      88.92      |
> | FedET      |   89.78   | FedET           |      87.17      |
> | FedMD      |   89.68   | FedMD           |      86.07      |
> | reamFL+Avg |   90.25   | reamFL+Avg      |      88.13      |
> | reamFL+IoT |   91.00   | reamFL+IoT      |      88.05      |
> | CreamFL    | **94.99** | CreamFL         |    **92.43**    |
>
> We further change the server models from GRU and ResNet50 to BERT and ResNet101 to increase model complexity. From the table below, we observe that with larger server models, performances in both cases are improved, and CreamFL outperforms all baselines under both settings.
>
> | COCO+COCO  | | COCO+Flicker30K | |
> | :--------- | :-------: | :-------------- | :-------------: |
> | Methods    |  R@1_sum  | Methods         |     R@1_sum     |
> | FedGEMS    |   129.85   | FedGEMS         |      128.23      |
> | FedET      |   129.78   | FedET           |      129.11      |
> | FedMD      |   130.18   | FedMD           |      129.70      |
> | reamFL+Avg |   130.29   | reamFL+Avg      |      130.02      |
> | reamFL+IoT |   130.93   | reamFL+IoT      |      130.41|
> | CreamFL    | **133.16** | CreamFL         |    **132.88**    |
>
> **Practical and complex scenarios:**
>
> To simulate more complex scenarios, two additional datasets, CIFAR10 and YelpReviewPolarity, are introduced as private data. In this experiment, we divide clients into two groups. In one group, we replace their datasets (image/text/multimodal) with CIFAR10, YelpReviewPolarity and COCO (a different subset from public data), respectively, while keeping the total number of clients and the data size unchanged. In the other group, data and settings are unchanged. BERT and ResNet101 are used as server models to further simulate larger-scale scenario. Under this setup, CreamFL also achieves superior performance.
>
> | Methods    | R@1_sum |
> | :--------- | :-----: |
> | FedGEMS    | 129.80  |
> | FedET      | 130.05  |
> | FedMD      | 130.16  |
> | reamFL+Avg | 130.84  |
> | reamFL+IoT | 131.50  |
> | CreamFL    | **133.06**  |
>
> Regarding practicality, we follow previous FL works and have sampled data for clients in a non-iid manner within each dataset. More clarifications have been added to Section 4.1.
>
> **Q2: Communication costs of KD-based frameworks.**
>
> KD-based frameworks, such as FedMD, also transfer representations of public data, so they have identical communication costs (total bits per communication) under the same settings. FedAvg and FedIoT transfer model parameters instead of representations and their communication costs are dependent on their adopted model architecture. We studied the performance and communication trade-off of several baseline methods and these results are included in Appendix A.4.
>
> **Q3: Ablation results of the reamFL+LCR.**
>
> Thanks for your comments. We perform additional ablation studies of LCR on reamFL+Mean, where mean aggregation is used as the baseline aggregation strategy. Results below show that using LCR for regularizing local training yields better performance than the reamFL+Mean setting, further demonstrating the effectiveness of LCR regularization.
> | Methods                 | R@1_sum |
> | :---------------------- | :-----: |
> | reamFL+Mean             |  85.75  |
> | reamFL+Mean + LCR.inter |  87.37  |
> | reamFL+Mean + LCR.intra |  86.94  |
> | reamFL+Mean + LCR       |  **88.57**  |

---

> ### Author Response · Authors · 2022-11-30
> **Follow-up. Thanks for the feedback again.**
>
> We thank the reviewer again for the supportive review. Additional experiments have been included, please let us know if our corrections, clarifications and additional results have addressed your concerns. We would be happy to address further concerns or questions.
>
> Thanks!

---

### Decision · Program_Chairs · 2023-01-20

**Decision:**

Accept: poster

**Justification For Why Not Higher Score:**

The ideas of using knowledge distillation and contrastive learning for federated learning has been sufficiently explored in recent years.

**Justification For Why Not Lower Score:**

The paper studies an important problem and presents a new solution for KD-based multi-modal federated learning. The paper is well written, and the extensive experiments demonstrate the advantages of the proposed method over baselines.

**Metareview: Summary, Strengths And Weaknesses:**

This paper presents a multimodal federated learning framework, named Contrastive Representation Ensemble and Aggregation for Multimodal FL (CreamFL), to handle the heterogeneous model architectures and data modalities. In particular, a global-local cross-model ensemble strategy is proposed to fuse multimodal representations. Experimental results on multiple real-world datasets are reported and discussed.

Overall, this paper is well written and easy to follow. The authors designed a novel framework to deal with multimodal discrepancy, i.e., modality gap and task gap. In addition, the authors provided a comprehensive overview of related work. The proposed knowledge ensemble transfer schema is able to tackle the challenge that the client owns different model architectures. However, during the discussion phase, there are still some remaining concerns, such as the use of public dataset, which should be further clarified in the final version of this work.

**Note From Pc:**

if the above contains the word "oral" or "spotlight" please see: "oral" presentation means -> notable-top-5% and "spotlight" means -> notable-top-25%. As stated in our emails, we are disassociating presentation type from AC recommendations

**Summary Of Ac-Reviewer Meeting:**

During the AC-reviewer meeting, reviewers confirmed that some of their concerns have been addressed by the authors' responses, such as the clarifications on model design and technical details. However, reviewers still concerned about the use of public datasets. The authors should clearly justify it in the final version.